# Effect of Treatment Adherence Improvement Program in Hemodialysis Patients: A Systematic Review and Meta-Analysis

**DOI:** 10.3390/ijerph191811657

**Published:** 2022-09-15

**Authors:** Hana Kim, I. Seul Jeong, Mi-Kyoung Cho

**Affiliations:** Department of Nursing Science, School of Medicine, Chungbuk National University, Cheongju 28644, Korea

**Keywords:** hemodialysis, treatment adherence and compliance, meta-analysis, systematic review

## Abstract

Herein, we performed a meta-analysis evaluating the effects of treatment adherence enhancement programs on treatment adherence and secondary outcomes for hemodialysis patients. Twenty-five Korean and international articles published prior to 31 March 2022 were selected following the PRISMA and Cochrane Systematic Review guidelines. We calculated summary effect sizes, conducted homogeneity and heterogeneity testing, constructed a funnel plot, and performed Egger’s regression test, Begg’s test, trim-and-fill method, subgroup analyses, and univariate meta-regression. The overall effect of treatment adherence enhancement programs for hemodialysis patients was statistically significant (Hedges’ g = 1.10, 95% CI: 0.77, 1.43). On performing subgroup analysis to determine the cause of effect size heterogeneity, statistically significant moderating effects were found for a range of input variables (Asian countries, study centers, sample size, study design, intervention types, number of sessions, quality assessment scores, funding, and evidence-based interventions). On univariate meta-regression, larger synthesized effect sizes were found for a range of study characteristics (Asian populations, single-center studies, studies with <70 participants, quasi-experimental studies, educational interventions, studies with >12 sessions, studies with quality assessment scores above the mean, unfunded studies, and non-theory-based interventions). Our results provide evidence-based information for enhancing program efficacy when designing treatment adherence enhancement programs for hemodialysis patients.

## 1. Introduction

Hemodialysis has a higher dependence on hospital treatment than peritoneal dialysis, and thus treatment adherence is more important. However, poor treatment adherence is a common problem among hemodialysis patients [1], leading to acute and chronic complications and an increase in mortality and morbidity [2,3].

Meta-analysis has recently been suggested as an important and useful analytic method for evidence-based evaluations of program efficacy [4,5]. It provides holistic information on the effects and efficacy of the evaluated programs and allows for the comparison of similar programs and the identification of components impacting effect size [6]. In this study, we performed a meta-analysis of the effects of treatment adherence enhancement programs for hemodialysis patients to identify factors affecting program effect size.

The results of our analysis will provide treatment adherence improvement program managers with relevant, evidence-based information, which can increase program efficacy. Using a descriptive meta-analysis that provides information on the causality of treatment adherence enhancement programs in consideration of effect size as well as the results of descriptive analyses of effect size and control variables [7,8], this study provides evidence-based information on factors affecting program effect size. Moreover, given that the majority of adherence intervention studies enrolling dialysis patients applied a non-randomized design [9] and that it is a recent trend to include both randomized and non-randomized intervention studies in systematic reviews and meta-analyses [10,11,12], we included quasi-experimental studies.

## 2. Materials and Methods

### 2.1. Design

Herein, we conducted a systematic review and meta-analysis analyzing the effects of treatment adherence programs for hemodialysis patients to determine factors affecting program efficacy and explain their interrelationships.

### 2.2. Eligibility Criteria and Outcome Variables

The study was conducted following the Preferred Reporting Items for Systematic Reviews and Meta-Analyses (PRISMA) [13] and Cochrane Handbook for Systematic Reviews of Intervention [14] guidelines. A systematic literature search was conducted to inform article selection based on the PICO-SD (Population, Intervention, Comparison, Outcome, Study Design) framework.

The targeted study population (P) was hemodialysis patients aged ≥18 years. Studies enrolling patients that were simultaneously enrolled in other studies (e.g., various clinical trials) deemed to affect treatment adherence were excluded. Intervention (I) indicates interventions designed to improve treatment adherence or compliance. For the present investigation, eligible studies were those that administered interventions to improve adherence to at least one of the following treatment categories: diet, fluid, medication, and dialysis treatment.

The comparison (C) group consisted of patients receiving general treatment. Regarding outcome (O) variables, the primary outcome measure was defined as self-reported treatment adherence or self-reported treatment compliance. Secondary outcome measures were as follows: interdialytic weight gain (IDWG), serum phosphorus levels (P), and serum potassium levels (K). Studies that did not present the results of self-reported treatment adherence or compliance and those that measured treatment adherence in terms of treatment non-compliance or non-adherence were excluded from the analysis.

The eligible study designs were randomized controlled trials (RCTs) and quasi-experiments. Among the quasi-experimental studies, single-group comparative studies were excluded.

The inclusion and exclusion criteria (Table 1) were applied to the literature search in both domestic and international electronic databases. Selected articles were those written in English or Korean as well as those reporting on the efficacy of interventions for enhancing treatment adherence in terms of means, standard deviations, and sample sizes.

### 2.3. Search Strategies

Using eight electronic databases (five international [PubMed, Embase, CINAHL, the Cochrane Library, and Scopus] and three Korean [Research Information Sharing Service (RISS), KMbase, and KoreaMed] databases) and applying the eligibility criteria (Table 1), articles published from database inception through 31 March 2022 were collected. The search period was from 1 December 2021 to 31 March 2022. The search protocol was registered on the PROSPERO International Prospective Register of Systematic Reviews (registration no. CRD42022347841, available at: https://www.crd.york.ac.uk/prospero/#searchadvanced, accessed on 11 August 2022).

Keywords including hemodialysis, treatment adherence, and treatment compliance, selected based on the PICO model, were verified within the MeSH database provided by PubMed before the search. Search terms were adjusted to suit each database, using MeSH terms and text words as appropriate. Moreover, the reference lists of the included studies and those of other important systematic and narrative reviews on relevant topics were hand-searched, and articles that may have been overlooked in previous searches were thereby identified.

The literature selection process consisted of four determination steps: identification, screening, eligibility, and total included studies (as specified in the PRISMA Statement). After removing duplicates, the inclusion and exclusion criteria were applied in the article screening process.

### 2.4. Quality Assessment

The checklists for RCTs and quasi-experimental studies provided in the Joanna Briggs Institute of Critical Appraisal Tool [15] were used for the quality assessment of the selected articles. For RCT quality assessment, 13 binary items (0 = no/unclear, 1 = yes) were used, with the total score ranging between 0–13 points. Quality assessment of the quasi-experimental studies was conducted using nine binary items (0 = no/unclear, 1 = yes), with the total score ranging between 0 and 9 points.

Two researchers (HNK and ISJ) independently conducted the quality assessment using checklists. Moreover, before assessment, two pilot tests were conducted for each study type to check the agreement between scoring for the two researchers. As a result of the pilot tests, different opinions were expressed on two items each from the RCT (No. 4: participant blinding, and No. 11: reliability in determining outcome measures) and quasi-experimental study checklist (No. 3: exposure to similar treatment, and No. 5: multiple measures). The two researchers discussed the contents of the articles concerned with the aid of the checklist manuals until all disagreements were resolved and an overall consensus was reached. Additional pilot tests were conducted using two additional articles, and the derived scores showed good interrater agreement.

### 2.5. Data Collection

Throughout the entire data collection and screening process, all articles selected for analysis were separately reviewed by three independent researchers (MKC, HNK, and ISJ). First, among the articles identified in the database search, duplicates were removed using Excel software. Additional articles were removed while the researchers were performing title and abstract screening based on the inclusion and exclusion criteria. Finally, full-text screening applying the specified inclusion and exclusion criteria was performed for the final article selection. Serial numbers were assigned to the selected articles according to bibliographic information, which was then placed in folders.

Researchers recorded the reasons for article exclusion and adjusted their respective opinions on article inclusion and exclusion before confirming the quality assessment results. Disagreements were resolved by discussion among the three researchers, who determined the final article selection through a consensus process. All disagreements were resolved by discussion during the screening process.

Serial numbers were reassigned to the articles that were selected for analysis. Information on author(s), year of publication, country of publication, number of study centers, number of participants, participant characteristics, study types, the types and characteristics of the evaluated interventions, outcome variables, quality assessment scores, funding, and the presence of theory-based interventions were extracted and coded. Theory-based intervention means that the intervention program is constructed based on theory.

### 2.6. Data Analysis

Study characteristics were described using means, standard deviations, and sample sizes. The coded data were analyzed using MIX 2.0 (i.e., professional software for meta-analysis in Excel) version 2.016 (BiostatXL, Mountain View, CA, USA) statistical software.

Study homogeneity in terms of effect size was tested according to the null hypothesis within chi-square testing by calculating the Cochrane *Q* value, and study heterogeneity was tested by calculating Higgin’s I^2^ and T^2^ values. In the case of this study, meta-analysis was performed using a random effect model since inter-study heterogeneity was confirmed and given the diverse characteristics of the evaluated studies. For a comparison among the studies, the effect size, which is a standardized form that can be compared, was used. The effect size of each study was calculated using the data measured immediately after the intervention of the experimental and the control group. The effect size is presented as Hedge’s g, based on the mean since Cohen’s d tends to overestimate the effect size when the sample is small [4]. Cohen’s d is the standardized mean difference between groups divided by the integrated standard deviation, and Hedge’s g is the adjusted effect size calculated by multiplying Cohen’s d by the correction factor [16]. If there are multiple intervention groups in the same study, the effect size was calculated for each data analysis as an independent study.

Publication bias, which refers to a tendency towards publishing only when the intervention effect is large and statistically significant, is routinely assessed to test the validity of studies within a meta-analysis that synthesizes and analyzes information on individual studies [14]. To assess the publication bias in the present study, visual analysis was performed using funnel plots and d values and was evaluated using Egger’s regression and Begg’s test.

To identify the causes of inter-study heterogeneity, subgroup analyses were performed based on study characteristics (country of publication, number of study centers, number of participants, study design, types and characteristics of the evaluated interventions, quality assessment scores, the presence of funding support, and the presence of theory-based interventions). In addition, for the meta-regression analysis, a univariate meta-regression was performed. In this methodology, effect sizes and correlations are assessed using only one moderating variable for analysis; all the moderating variables used in the subgroup analysis are input variables. Every analytic process applied to the meta-regression analysis methodology was carried out using a fixed effects model and the method of the moment to estimate the variance, 95% confidence interval (CI), two-sided *p*-value, and *Z*-value.

## 3. Results

### 3.1. Data Extraction

A total of 4905 articles were retrieved in the database search (1236 from PubMed, 345 from Embase, 239 from CINAHL, 619 from Cochrane Library, 2384 from Scopus, and 82 from RISS). After removing duplicates and a manual review by the researchers, the remaining 3245 articles underwent more comprehensive screening for eligibility. Finally, 25 articles were selected. Of these, the studies by [17,18] included two experimental groups (a, b), and those by [19,20] included three experimental groups (a, b, c) (Figure 1).

### 3.2. Study Characteristics

Table 2 outlines the characteristics of the studies evaluated in the present meta-analysis. These studies were published between 2006 and 2021, with the largest proportion of the evaluated studies (*n* = 11.44%) published during the last five years (2017–2021). When classified by country of publication, we determined that twelve studies were published in East Asia (Korea or Singapore), eight studies were published in West Asia (Iran or Turkey), two studies were published in South Asia (India), and three studies were published in non-Asian countries (Greece, UK, and the USA). The sample sizes of the included studies ranged between 40 and 235 participants (median: 70 participants). Studies with a sample size above the median (*n* = 14.56%) outnumbered studies with a sample size below the median (*n* = 11.44%).

With regards to study design, RCTs (*n* = 13.52%, 1250 participants in total) slightly outnumbered quasi-experimental studies (*n* = 12.28%, 775 participants in total). The classification by intervention type was as follows: educational programs (*n* = 20.80%), self-management programs (*n* = 1), self-efficacy programs (*n* = 2), and “other” (*n* = 2: acupressure and motivational interviewing).

Regarding the intervention structure, nineteen (76%) studies evaluated programs administering 12 or fewer sessions, four studies (16%) evaluated programs exceeding 12 sessions, and two studies [21,22] and one study group were evaluated in the study by [17] (E1: the pamphlet intervention) reported on 1 session only. The program duration per session ranged between 5 and 120 min, with a mean duration of 38 min. Ten studies (40%) were funded research. Theory-based interventions were administered in six studies (24%).

In terms of outcome measures, all 25 studies evaluated treatment adherence (the primary outcome measure) using assessment tools such as the End-Stage Renal Disease Adherence Questionnaire (ESRD-AQ), the Greek version of the Simplified Medication Adherence Questionnaire for Patients undergoing Hemodialysis (GR-SMAQ-HD), the Fluid Control in Hemodialysis Patients Scale (FCHPS), the Medical Outcomes Study (MOS) scale, the eight-item Morisky Medication Adherence Scale (MMAS-8), and the four-item Morisky Medication Adherence Scale (MMAS-4). The ESRD-AQ was the tool used in the largest number of studies (*n* = 6).

Among the secondary outcome measures, IDWG = P and K was assessed in 13, 10, and 9 studies, respectively.

### 3.3. Methodological Quality

To ensure interrater concordance on the quality assessment score before assessment, four pilot tests (two RCTs and two quasi-experimental studies) were conducted; this resulted in a concordance rate of 79.55%. The mean quality score was 8.23 (range: 6–11; maximum score: 13) for the 13 RCTs and 7.58 (range: 6–8; maximum score: 9) for the 12 quasi-experimental studies.

Out of 13 RCT checklist items, five items (random assignment, follow-up completion, intention-to-treat analysis, similarity in assessing outcome measures, and appropriate trial design) were clearly explained in all 13 RCTs. The items “blinding of delivering treatment” and “reliability in outcome measures” were clearly explained in only one RCT, and the item “blinding of the outcome assessor” was only assessed in two RCTs.

Out of nine checklist items for quasi-experimental studies, five items (clarity of the cause-and-outcome effect, comparison of the treated groups, multiple measures, similarity in outcome measurement, and appropriate statistical analysis) were clearly explained in all 12 quasi-experimental studies. The item “reliability in outcome measurement” was clearly explained in eight studies, whereas the item “exposure to similar treatment” was only described in two studies.

A total of 25 studies were selected for this review. These studies obtained quality scores above the median of total score and thus met the established literature selection criteria for systematic reviews of quantitative studies [41] (Table 3).

### 3.4. Effects of Intervention Programs on the Primary Outcome Measure

In the 25 selected studies, the adjusted standardized mean difference (Hedges’ g) evaluating treatment adherence between the treatment and control groups was calculated using means, standard deviations, and sample sizes, as presented in the synthesis forest plot (Figure 2). The analysis revealed that treatment adherence was significantly increased post-intervention according to the random effects model (Z = 6.63, *p* < 0.001), with the overall program effect size estimated at 1.10 (fixed effect model: 0.78); this far exceeded the cut-off value of 0.8 [42]. The presence of inter-study variance was confirmed given T^2^ = 0.76 (95% CI: 0.57, 1.00) and Q = 350.67 (Q-*df* = 318.68, *p* < 0.001). Moreover, with I^2^ = 91.0%, a considerable level of heterogeneity in effect size was confirmed. Accordingly, a subgroup analysis was performed to derive an exploratory explanation of study heterogeneity.

The analysis revealed that, out of nine analysis variables (country of publication, number of study centers, number of participants, study design, types of interventions, number of sessions, quality assessment scores, the presence of funding support, and the presence of theory-based interventions), all studies except those conducted in non-Asian countries (Greece, the UK, and the US) showed moderating effects on the effect size for treatment adherence in all subgroups (Table 4). Whereas no significant effects of treatment adherence enhancement programs were found in the three non-Asian studies (Z = 0.92, *p* = 0.360), significantly positive effects were confirmed in studies published in East Asia (Z = 4.10, *p* < 0.001), West Asia (Z = 4.37, *p* < 0.001), and South Asia (Z = 4.02, *p* < 0.001), with respective effect sizes (ES) of 0.85 (95% CI: 0.45, 1.26), 2.10 (95% CI: 1.16, 3.05), and 0.90 (95% CI: 0.46, 1.32).

Regarding the number of study centers, multicenter studies showed a medium ES (0.71, 95% CI: 0.35, 1.07) and single-center studies showed a large ES (1.35, 95% CI: 0.87, 1.82). Regarding the number of participants, taking the median value of 70 as a reference, both below-median (ES = 1.12, 95% CI: 0.69, 1.56), and above-median (ES = 1.10, 95% CI: 0.67, 1.53) studies showed a large ES, as did studies with both quasi-experimental (ES = 0.97, 95% CI: 0.61, 1.33) and RCT (ES = 1.27, 95% CI: 0.75, 1.80) designs. Moreover, while non-educational programs showed a medium ES (0.66, 95% CI: 0.20, 1.12), educational programs showed a large ES (1.23, 95% CI: 0.83, 1.63). Large ES values were also evident for programs with >12 sessions (ES = 2.94, 95% CI: 1.36, 4.51) and quality scores above the mean (RCTs: 8.23, quasi-experimental studies; 7.58) (ES = 1.34, 95% CI: 0.85, 1.82). Unfunded studies showed a large ES (1.51, 95% CI: 0.97, 2.05), whereas funded studies showed a medium ES (0.68, 95% CI: 0.37, 0.98). Studies evaluating theory-based programs showed a medium ES (0.70, 95% CI: 0.11, 1.29), and those evaluating non-theory-based programs showed a large ES (1.23, 95% CI: 0.84, 1.62).

Meta-regression analysis was performed to investigate the possibility of further explaining heterogeneity according to differences in study characteristics or the evaluated study population (Table 5). Univariate meta-regression revealed that all variables showed statistically significant effects. Regarding the relationships between individual variables and treatment adherence effect size (i.e., the dependent variable; Hedges’ g = 1.10), more positive effects were exerted on the effect size for the evaluated treatment adherence enhancement programs in studies conducted in Asian vs. non-Asian regions (Z = 0.74, *p* < 0.001), single-center vs. multicenter studies (Z = 4.92, *p* < 0.001), studies with smaller enrolled populations (<70 vs. ≥70; Z = 3.90, *p* < 0.001), quasi-experimental studies vs. RCTs (Z = −2.22, *p* = 0.026), studies conducting educational vs. other interventions (Z = 3.52, *p* < 0.001), studies with a higher number of administered sessions (≥12 vs. <12; Z = 3.99, *p* < 0.001), studies with quality assessment scores above vs. below the mean (Z = 2.43, *p* = 0.015), unfunded vs. funded studies (Z= −5.06, *p* < 0.001), and studies evaluating non-theory-based vs. theory-based programs (Z= −4.84, *p* < 0.001).

### 3.5. Effects of Intervention Programs on Secondary Outcome Measures

IDWG, P, and K were evaluated as the secondary outcome measures in the present meta-analysis. IDWG, P, and K were measured as the outcome variables in 13, 10, and 9 studies, respectively (out of the 25 studies selected for analysis). The overall IDWG-related program effect size was small (−0.29 [95% CI: −0.52, −0.06]), with a statistically significant post-interventional decrease in IDWG (Z = −2.48, *p* = 0.013). The overall effects of the program on P (Z = −1.38, *p* = 0.170) and K levels (Z = −1.20, *p* = 0.230) were not statistically significant (Table 6).

### 3.6. Publication Bias

Publication bias was examined using a funnel plot analysis, and the symmetry of the article dispersion around the mean effect size was visually inspected (Figure 3). We found that the individual effect sizes of the 25 papers were asymmetrically dispersed on the upper left and lower right sides of the funnel plot, indicating some publication bias. Therefore, Egger’s regression test was additionally performed for asymmetry analysis, and the significance probability of the intercept of the regression analysis was found to be statistically significant (*p* < 0.001). Likewise, Begg’s test for rank correlation showed a tau-b correlation of 0.40 and 31 ties (*p* < 0.001).

With publication bias thus confirmed by both test results (Table 7), the trim-and-fill method [43] was used to illustrate the effects of publication bias on the study results (Figure 3, Table 7). The trim-and-fill method allows for estimating the number of missing or unreported studies and the ensuing effects as well as comparing the differences between original and corrected numbers of articles and their effect sizes. Using the trim-and-fill method, the number of articles to be corrected and added to the 25 articles originally included in this study was estimated as 8, and the effect sizes of the original 25 papers and the corrected 33 papers were estimated as 1.10 and 0.50, respectively. Despite a decrease in the effect size of treatment adherence from large (before correction) to medium (after correction), both pre- and post-correction effect sizes were confirmed as statistically significant. Therefore, despite the presence of publication bias in this study, our follow-up evaluations indicate that it did not affect the analytic results for the synthesized effect size when evaluating treatment adherence, confirming the acceptability of the effect size estimated in this meta-analysis.

## 4. Discussion

The purpose of this study was to analyze the effect size for treatment adherence enhancement programs and identify factors affecting effect sizes through a meta-analysis of treatment adherence enhancement programs for hemodialysis patients. This study evaluated effect sizes and subgroup characteristics for moderating variables of the 25 selected studies, which were conducted to improve treatment adherence among hemodialysis patients. In addition, univariate meta-regression analysis was used to analyze the characteristics of the moderating variables causing the differences in effect size for the treatment adherence enhancement programs, and the predictive variables affecting the treatment compliance effect size were explained along with the characteristics of the predictive model.

The results of this study confirmed the effectiveness of the individual programs that have been implemented to improve treatment adherence for hemodialysis patients. The overall effect size of the programs was large (1.10) when estimated using a random effects model (fixed effects model: 0.78), and treatment adherence significantly increased post-intervention. These findings are consistent with a systematic review and meta-analysis of RCTs conducted to improve treatment adherence (diet, fluid intake, dialysis, and medication) [44], a meta-analysis evaluating psychosocial and educational interventions for enhancing treatment adherence [45], a meta-analysis evaluating educational and self-management interventions for enhancing adherence to dialysis [46], a meta-analysis of nursing interventions for improving treatment adherence [47], and a systematic review of interventions for improving hemodialysis adherence [48]. These findings corroborate the positive effects for treatment adherence enhancement programs evaluated in a previous meta-analysis of dietary educational interventions for managing hyperphosphatemia in hemodialysis patients [11]. Likewise, in a study evaluating the effects of self-care interventions on IDWG, a widely used surrogate endpoint of treatment adherence, IDWG significantly decreased after the intervention; this is consistent with the results of a present study [10]. However, there were certain differences in the assessment of the primary study outcomes. Whereas the current study evaluated program efficacy by calculating the effect size as derived from 25 studies examining self-reported treatment adherence, other studies have used objective indicators (i.e., surrogate measures such as IDWG, serum K, and serum P) following WHO and Kidney Disease Outcomes Quality Initiative (KDOQI) guidelines [45,47].

Treatment adherence can be assessed using direct measurement methods, such as documenting the number of medications used, establishing a medication event monitoring system (MEMS) [49], and documenting participation in dialysis sessions. However, since these methods do not lend themselves well to quantification, surrogate measurement methods, such as monitoring IDWG and blood testing, are used to evaluate treatment adherence [44]. However, these surrogate measurement results may be impacted by factors other than treatment adherence, including residual renal function, the quality of dialysis, and improper dialysis procedures [9]. Moreover, given concerns about overestimation when evaluating self-reported adherence, caution is warranted in interpreting study findings [9]. Nevertheless, given the fact that results derived through surrogate measurements (objective indicators) and the self-reported (subjective) measures evaluated in the current study coincide, it may be assumed that the assessment method used in this study is a viable method for assessing program performance.

Assessment based on self-report can also be a good assessment method. More specifically, a surrogate measurement made while evaluating participation in the administered program can be influenced by factors other than the patient’s efforts, as mentioned previously, and can result in underestimation incommensurate with the patient’s efforts. This may in turn lead to reducing motivation for program participation.

A significant level of study heterogeneity was detected herein. This finding is attributable to the use of varied instruments (ESRD-AQ, GR-SMAQ-HD, FCHPS, MOS, MMAS-8, MMAS-4), including in-house instruments, to measure treatment adherence (the primary outcome measure), as well as to differences in study characteristics (e.g., the evaluated studies administered a widely dissimilar number of intervention sessions; range: 1–24 sessions). This heterogeneity may pose a problem on the potentially limited reliability of the results of the present investigation. Therefore, subgroup analysis and meta-regression analysis were performed to identify the cause of heterogeneity. The subgroup analysis revealed that the effect size for treatment adherence was significantly affected by all of the evaluated moderating variables, including country of publication (except for the US, the UK, and Greece), number of study centers, number of study participants, study design, intervention type, number of administered sessions, quality assessment scores, funding (yes/no), and theory-based interventions (yes/no). However, caution is warranted when interpreting these results given the lower number of studies included in this sub-analysis, thereby inevitably decreasing statistical power and increasing the likelihood of false positive or false negative findings [50]. Caution is also warranted on drawing causal inferences, although the detected findings may be considered for evaluating causation on differences in treatment adherence, and these considerations may serve as a basis for proposing new hypotheses for follow-up research, which allows us to draw causal inferences [50].

Moreover, univariate meta-regression (performed using the same variables as those specified above) confirmed the significantly moderating effects of all input variables on the effect size for treatment adherence. Although the studies published in Asia showed significantly increased treatment adherence, relative to those published in non-Asian countries (Greece, the US, and the UK), caution is warranted when interpreting these analytic results given the risk of distorted interpretation due to the sparsity of the non-Asian studies included herein (*n* = 3). Larger effect sizes were observed in single center vs. multi-center studies and in studies with <70 participants vs. those with ≥70 participants. These findings are consistent with the result of a previous study comparing the effect sizes derived within RCTs [51]. The finding of single-center studies showing larger effect sizes as compared to multi-center studies may be explained by the effect of small-scale research and by the nature of meta-analysis. For example, small-scale studies tend to have greater efficacy. This tendency was also confirmed in this investigation, wherein all studies with <70 participants were single-center studies.

Regarding intervention type, we found that educational programs presented larger effect sizes than other intervention programs (i.e., self-management programs, self-efficacy programs, acupressure, and motivational interviewing), which is consistent with the findings of previous meta-analyses that verified the positive effects of educational programs on improving treatment adherence [10,45,46]. Educational interventions are frequently used to develop healthy behaviors in hemodialysis patients, and these interventions are designed to help patients adapt to disease, treatment, and behavioral changes by informing them about their health needs and conditions [46]. In previous systematic literature reviews and meta-analyses [11], half of the included studies demonstrated statistically significant improvements in treatment adherence after just one education session, showing the high level of applied information processing demonstrated by hemodialysis patients. In contrast, some prior studies have indicated that education-mediated information provision is not sufficient to change behavior [52,53] and that behavior change should instead be induced through self-management interventions that go beyond information transfer; we note that no difference was observed between self-management and educational interventions in a previous meta-analysis [46] and that this finding is supported by the results of the present meta-analysis. Distinct from various other programs (such as drug therapy), educational intervention presents a low-risk option as it has no health-related side effects and has the documented effect of strengthening willingness to adhere to treatment [54].

Regarding the number of intervention sessions, we found that the effect size for treatment adherence was larger in studies with >12 sessions than in those with ≤12 sessions. Considering that intervention-induced compliant behavior is likely to revert to noncompliant behavior, a recurrence prevention strategy needs to be established to ensure long-term compliance [52]. Thus, repeated sessions are considered to have contributed to enhancing treatment adherence in the present investigation. In addition, four of the evaluated studies providing >12 sessions were educational interventions that were confirmed as having large effect sizes within the current analysis. Of these four studies, three obtained quality scores that were above the mean value, suggesting that study characteristics can influence effect size estimates.

Moreover, we found that studies with higher quality scores demonstrated higher treatment adherence. The quality of a study refers to the extent to which the study measures “real” effects (i.e., the study’s validity) [55]. Moreover, the validity of a study has two dimensions: (i) internal validity, which represents the extent to which study results accurately reflect the research situation (e.g., the extent to which the study minimizes biases), and (ii) external validity, which presents the extent to which the results of a study can be generalized [55]. In conducting a meta-analysis, the quality assessment of a study is a validity assessment; that is, a study that has obtained a high-quality score is a study that accurately reflects the research situation and has high generalizability, and such studies are advantageous in deriving accurate, positive results. Concomitantly, the univariate meta-regression conducted in this study revealed that RCTs showed smaller effect sizes than quasi-experimental studies, presumably due to differences in the quality of the study design, with RCTs demonstrating 63.30% of the maximum quality score (8.23 out of 13 points) vs. 84.22% (7.58 out of 9 points) among quasi-experimental studies.

Concerning funding, funded studies were found to have smaller effect sizes than unfunded studies. Funded studies can easily afford to provide a strong incentive as compensation for participating in the study, in contrast to unfunded studies in which researchers themselves are accountable for providing compensation and cannot afford as much compensation as funded research. In the latter case, larger effect size can be obtained because the enrollment patients decide to participate in the program with a true intention to participate due to the health-promoting effects that are expected due to participating in the program, rather than for monetary compensation. Moreover, in a previous study, physicians’ recommendations and health expectations were found to be important considerations on the intention and decision to participate in a clinical trial regardless of financial compensation [56]. In contrast, for those participating in a clinical trial with the expectation of financial compensation, the level of hassle and burden related to the participation process (such as the number of required blood samples and related tests, participation frequency, associated risks, and study duration), in contrast to expectations about health-promoting effects, were the primary determinants of participation [56]. From this, it can be inferred that program logistics had a smaller effect on these latter participants.

Behavioral change is a complex phenomenon, and a program based on a theoretical framework can administer systematic interventions [46]. According to the results of the present investigation, studies conducted in the absence of theoretical frameworks were found to have larger effect sizes than theory-based programs. This finding is partially consistent with the results of a previous meta-analysis reporting that, in a study employing Leventhal’s self-regulation theory, motivational counseling, and social cognitive theory, only Leventhal’s self-regulation theory had a positive effect on treatment adherence [11]. In addition, of the six evaluated studies that were conducted based on a theoretical framework, four were multi-center studies and five were studies with >70 participants. These findings allow for the assumption that differences in study results may be attributable to differences in study characteristics.

## 5. Limitations

The results of the present systematic review and meta-analysis provide knowledge regarding the intervention types that are more effective in enhancing treatment adherence as well as overall evidence-based information for healthcare professionals managing interventions aiming toward treatment adherence enhancement in hemodialysis patients. However, we acknowledge some limitations of the current investigation. For example, some studies did not provide detailed explanations and information about the implemented interventions, including intervention methods, duration, as well as number and frequency of sessions. Moreover, of the 25 selected studies, 8 did not perform homogeneity testing and may have generated exaggerated or biased results. In addition, the number of studies included in the subgroup analysis was small in certain categories. Given a higher risk of distorted results when a smaller number of studies is analyzed, it is necessary to add more studies for analysis in future research. Moreover, whereas a higher level of generalizability may be expected in studies conducted in diverse clinical settings, the use of several measurement tools (especially questionnaires developed for specific studies) is likely to lower the quality of the obtained measurement results due to a lack of testing for sample adequacy. Limiting the search languages and the literature databases to Korean and English sources may have led to difficulty in controlling the likelihood of biases on article selection. It is expected that follow-up studies will compensate for these limitations and contribute to additionally enhancing treatment adherence in hemodialysis patients.

## 6. Conclusions

In this study, a meta-analysis was performed to systematically and comprehensively verify the effectiveness of the treatment compliance promotion program for hemodialysis patients. As a result of selecting and reviewing 25 studies, it was confirmed that the program performed to improve treatment adherence was effective in enhancing treatment adherence. Based on the meta-regression analysis, studies conducted in Asia, single-center studies and studies with less than 70 participants, quasi-experimental studies, educational programs, sessions exceeding 12 sessions, above mean quality assessment score, non-funded studies, and programs that are not based on theory have a significantly larger effect size on treatment adherence.

As the primary outcome of this study, the previous study [44,45,47,49] used the results of direct or indirect measurement of treatment adherence, but there was a difference in the evaluation of the patient’s self-reported treatment adherence. Surrogate measurement results may be changed by factors other than adherence, and if improved results are not obtained despite the patient’s efforts, it may be underestimated and the motivation for adherence may be reduced. It was confirmed that the effect of the program measured through the subjective report and the results of the previous study derived through the surrogate measurement result were consistent, and thus evaluation through the subjective report could be a good method of evaluation.

The results of this study can be used as evidence to explain the effect of the compliance promotion program on the self-reported treatment compliance and IDWG, P, and K of hemodialysis patients. Therefore, it will be possible to use this information as useful evidence-based information to improve program performance in both design and execution when constructing a program to promote adherence to hemodialysis patients in the future.

## Figures and Tables

**Figure 1 ijerph-19-11657-f001:**
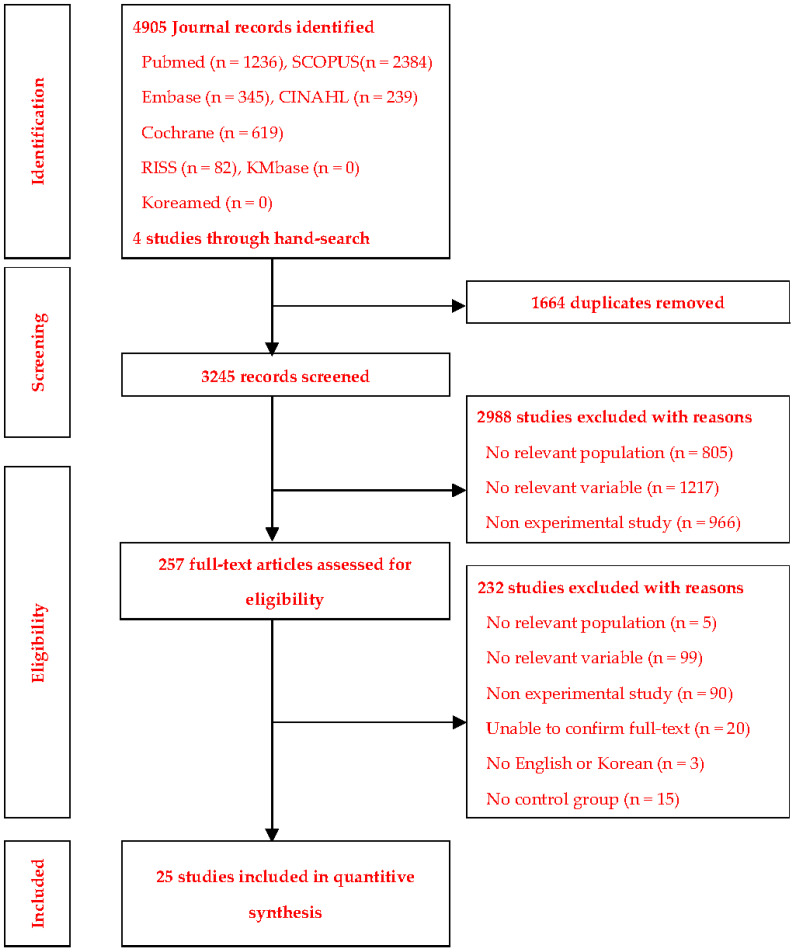
Study extraction process.

**Figure 2 ijerph-19-11657-f002:**
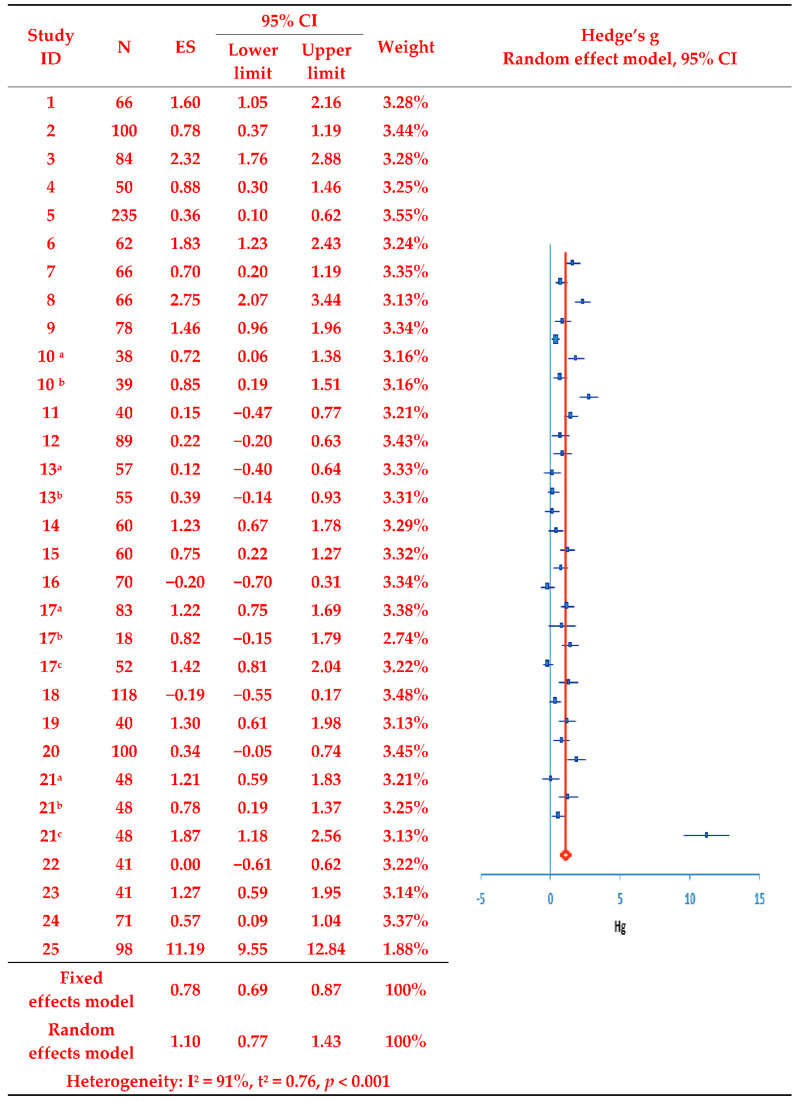
Effects of the evaluated intervention programs on treatment adherence. Duplicate removal of the number of participants in the study of [17,18,19,20]. Studies with two or more experimental groups are marked with a, b, and c for classification. N, number of subjects; ES, effect size; CI, Confidence interval; Hg, Hedge’s g.

**Figure 3 ijerph-19-11657-f003:**
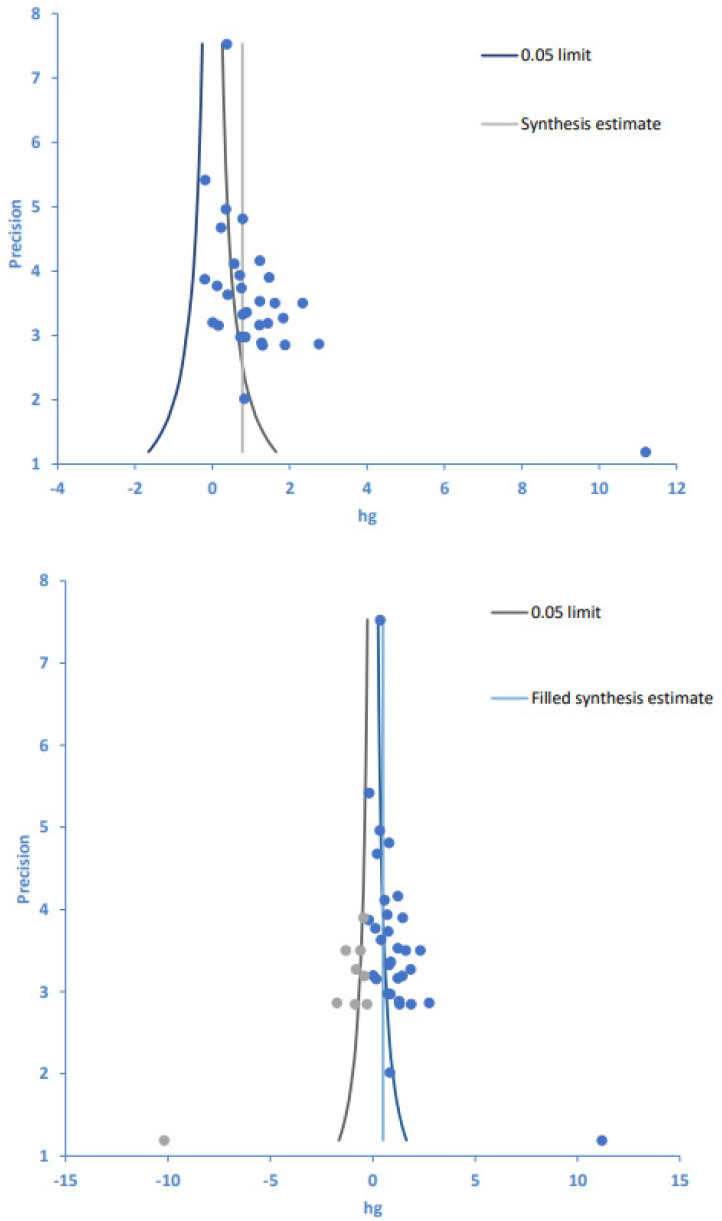
Funnel plot (**top**) and trim-and-fill plot (**bottom**) of intervention programs and treatment adherence. Blue circles are original data, gray circles are imputed filled values. Hg, Hedges’ g.

**Table 1 ijerph-19-11657-t001:** Study eligibility criteria.

	Inclusion Criteria	Exclusion Criteria
Participants	Adults aged >18 years on hemodialysis for chronic renal failure.	Enrolled study participants participated in other studies that may affect treatment adherence.
	Studies published through 31 March 2022.	
Intervention	Intervention programs related to treatment compliance or treatment adherence (medication, fluid, diet, or dialysis treatment).	---
	Studies report means, standard deviations, and concrete sample sizes.	
Control	Usual or standard treatment, comparative experiments.	---
Outcomes	Primary outcome: treatment compliance or treatment adherence indirectly measured through self-report questionnaires. When serial interventions were performed, only the effects measured at the end of the intervention were used for analysis.	Studies that did not measure treatment adherence or treatment compliance.
	Secondary outcomes: inter-dialytic weight gain, serum phosphorus levels, and serum potassium levels. When serial interventions were performed, only effects measured at the end of the intervention period were used for analysis.	Studies that measured non-compliance or treatment non-adherence.
Study design	Quasi-experimental studies or RCTs.	Non-quasi-experimental studies or non-RCTs.
		For quasi-experimental studies, single-group comparative studies were excluded.

Abbreviations: RCTs, randomized controlled trials.

**Table 2 ijerph-19-11657-t002:** Descriptive summary of the included studies.

No	Author	Year	Country	Center	Participants	Participants Characteristics	Design	Intervention	Characteristics of Interventions	Outcome Variables	Quality Score	Funding	Theory-Based
1	Vardanjani et al. [23]	2015	Iran	1	N = 66 (E = 33, C = 33)	Age ≥ 18 years Mean age (E = 54.0 [15.7], C = 53.4 [14.9]) HD ≥ 12 months ESRD-AQ ≤ moderate	RCT	Education program (individual, feedback)	8 sessions 1 h/session Researchers	Adherence (ESRD-AQ), IDWG, K, P, Na, Cr, BUN, Ca, Alb, alkaline phosphatase, Hb	8	Yes	No
2	Rodrigues et al. [21]	2019	India	1	N = 100 (E = 50, C = 50)	Age ≥ 18 years	RCT	Education & counseling program	Pharmacist	Adherence (medication adherence), medication knowledge	6	No	No
3	Park & Kim [24]	2019	Korea	1	N = 84 (E = 42, C = 42)	19 ≤ age ≤ 60 years Mean age (E = 51.5 [10.2], C = 48.9 [9.4]) HD ≥ 3 months Android-based smartphone user	Quasi-E	Integrated self-management program (face-to-face counseling and education (twice per month), use of mobile applications, questions about hemodialysis self-management)	4 sessions 8 weeks Researcher	Adherence, IDWG, K, P, self-efficacy, treatment	8	No	Yes (Orem’s self-care deficit theory, empowered caring model)
4	Alikari et al. [25]	2018	Greece	1	N = 50 (E = 25, C = 25)	18 ≤ age ≤ 65 years Mean age (E = 51.2 [11.5], C = 49.8 [8.5]) HD ≥ 3 months	Quasi-E	Educational intervention & booklet (Dialysis. Answers to common questions)	One-time personalized educational intervention (face-to-face) 45 min Researcher	Adherence (medication adherence: GR-SMAQ-HD), medication knowledge (KDQ), QoL (MVQoLI-15)	6	No	No
5	Griva et al. [26]	2018	Singapore	14	N = 235 (E = 134, C = 101)	Age ≥ 21 years Mean age (E = 53.1 [10.5], C = 53.9 [10.4]) HD ≥ 6 months	RCT	Self-management intervention (fluid intake, diet, medication)	4 sessions 120 min/session Researcher	Adherence (self-reported adherence), IDWG, K, P self-efficacy, self-management skills, Alb	11	Yes	Yes (social cognitive theory)
6	Yun & Choi [27]	2016	Korea	1	N = 62 (E = 30, C = 32)	Age ≥ 20 years Mean age (E = 61.2 [13.5], C = 54.7 [15.9]) HD ≥ 12 months	Quasi-E	Dietary self-efficacy program (diet record, video, individual & group, feedback)	8 sessions 20–65 min/session 8 weeks Researcher	Adherence (dietary adherence), IDWG, K, P, HRQoL, dietary QoL, Alb	8	No	Yes (self-efficacy theory)
7	Zhianfar et al. [28]	2020	Iran	1	N = 70 (E = 35, C = 35)	Age ≥ 18 years HD ≥ 3 months ESRD	RCT	Multifaceted educational intervention (education, CBT, social support)	8 sessions 90 min/session 8 weeks Researcher & psychiatrist	Adherence (ESRD-AQ), IDWG, nursing care satisfaction (PSNCQQ), QOL (WHOQOL-SF), perceived social support (MSPSS), depression (BDI-SF)	7	Yes	No
8	Arad et al. [29]	2021	Iran	1	N = 66 (E = 33, C = 33)	20 ≤ age ≤ 65 years Having a personal mobile phone and the ability to use it	RCT	Education program and nurse-led telephone follow-up (including SMS)	24 sessions (twice a week) 20 min/session 12 weeks Researcher	Adherence (ESRD-AQ), K, P, Na, Cr, BUN, Ca, Alb, Hb, iron, normalized protein catabolic rate (nPCR), and Kt/V	8	No	No
9	Baser & Mollaoglu [30]	2019	Turkey	1	N = 78 (E = 38, C = 40)	Age ≥ 18 years HD ≥ 12 months	Quasi-E	Education program (nutrition education booklet for dialysis patients)	4 sessions First session—20–25 min/session 2,3,4 sessions—10–15 min/session 16 weeks Researcher	Adherence (FCHPS), IDWG, DDFQ, target weight, UF volume, BP	8	No	No
10	Parvan et al. [17]	2015	Iran	1	N = 58 (E1 = 19, E2 = 20, C = 19)	Age ≥ 18 years Mean age (E1 = 47.2 [14.0], E2 = 50.5 [11.4], C = 61.4 [13.4]) HD ≥ 12 months	RCT	Education program (knowledge about the disease, medication, fluid restriction, diet)	E1—pamphlet, E2—2 consecutive sessions 20 min/session Researcher	Adherence (MOS), hemodialysis knowledge (CHeKS)	7	Yes	No
11	Kim et al. [31]	2015	Korea	1	N = 40 (E = 20, C = 20)	Mean age: 62.78 (10.64) years (E = 64.20 [9.47], C = 61.35 [11.77]) HD ≥ 12 months	Quasi-E	Education program (knowledge about the disease, exercise, medication, diet, weight control, pain, and sexuality)	8 sessions (once a week) 20~30 min/session 8 weeks Researcher	Adherence (compliance) IDWG, K, P, knowledge, Alb, Hb, Kt/V	8	Yes	No
12	Wileman et al. [22]	2016	UK	3	N = 89 (E = 49, C = 40)	Mean age (years) (E = 62.8 [14.9], C = 58.2 [16.0]) HD ≥ 3 months IDWG > 2.0 kg	RCT	Self-affirmation intervention (health information about Fluid control), adherence to treatments)	Nephrologist	Adherence, IDWG	10	No	Yes (self-affirmation theory)
13	Chang et al. [18]	2021	Korea	3	N = 84 (E1 = 29, E2 = 26, C = 28)	Age ≥ 20 years Mean age (E1 = 67.4 [11.4], E2 = 61.8 [14.1], C = 63.2 [14.9]) HD ≥ 6 months Available data for saliva measurements	Quasi-E	auricular acupressure and a fluid-restriction adherence program	6 sessions (once a week) 60 min/session 6 weeks Nurse	Adherence (fluid control), IDWG, DQOL, salivary flowrate	8	Yes	Yes (empowerment model)
14	Ok & Kutlu [32]	2021	Turkey	1	N = 60 (E = 30, C = 30)	18 ≤ age ≤ 65 years Mean age (E = 51.2 [11.5], C = 49.8 [8.5]) HD ≥ 6 months A patient who fulfills one or more criteria for treatment on adherence	RCT	Motivation interviewing (determine the cause of patient’s adherence problems, boost the motivation of change, discuss change, evaluate)	4 sessions 20~40 min/session 4 weeks Researcher	Adherence (ESRD-AQ), K, P, SF-36, daily weight gain or fluid intake, Alb, Kt/V	9	Yes	No
15	So et al. [33]	2006	Korea	1	N = 60 (E = 30, C = 30)	20 ≤ age ≤ 7 years HD ≥ 1 month	Quasi-E	Drug education program (knowledge, enhance medication adherence)	4 sessions 20 min/session 2 weeks	Adherence (medical compliance), medication knowledge	7	No	No
16	Lim et al. [34]	2018	Korea	1	N = 70 (E = 48, C = 22)	Age ≥ 18 years Mean age 58.9 (15.9) (E = 59.7 [16.4], C = 57.3 [14.9]) HD ≥ 12 months	RCT	Education (low-phosphate diet and phosphate binder intake)	One-time personalized educational intervention (face-to-face) 30 min Dietitians & pharmacist	Adherence (medication adherence; MMAS-8), P, number of patients who reached the goal of a calcium-phosphate product lower than 55, dietary phosphate intake, PG-SGA (phosphate intake, acknowledge of phosphate binder, the bioequivalent dosage of phosphate binder)	9	No	No
17	Mateti et al. [19]	2018	India	3	N(academic) = 83 (E = 4, C = 41) N(government) = 18 (E = 9, C = 9) N(corporate) = 52 (E = 27, C = 25)	18 ≤ age ≤ 75 year Mean age (academic) E = 52.78 (10.45), C = 49.40 (12.47) Mean age (government) E = 49.15 (12.57), C = 48.00 (17.00) Mean age E = 52.97 (15.12), C = 53.77 (11.87) HD ≥ 3 months	RCT	Pharmaceutical education & motivation intervention	52 weeks	Adherence (medication adherence; MMAS-8), IDWG, Hb, BP	9	No	No
18	Klein et al. [35]	2017	USA	6	N = 118 (E = 59, C = 59)	Age ≥ 18 years Mean age 58.9 (15.9) (E = 59.7 [16.4], C = 57.3 [14.9]) HD ≥ 6 months	RCT	Education & counseling program	12 sessions 10–15 min/session 12 weeks Nurse	Adherence (medication adherence; MMAS-4), IDWG, BP self-efficacy, BP control, sodium intake	7	Yes	Yes (self-regulation theory)
19	Kim & Yoo [36]	2006	Korea	1	N = 40 (E = 20, C = 20)	Age ≥ 20 years HD ≥ 6 months	Quasi-E	Education program	6 sessions 30 min/session 2 weeks Researcher	Adherence (compliance), K, P, knowledge, Cr, BUN, Alb	8	No	No
20	Kim & Han [37]	2016	Korea	1	N = 100 (E = 50, C = 50)	Age ≥ 19 years Mean age E = 57.30 (14.03), C = 58.52 (14.74) HD ≥ 1 month	Quasi-E	Education program (individualized diet education)	6 sessions 30 min/session 12 weeks Nurse	Adherence (compliance), knowledge	7	No	No
21	An [20]	2009	Korea	4	N = 96 (E1 = 24, E2 = 24, E3 = 24, C = 24)	Age ≤ 70 years HD ≥ 1 month	Quasi-E	Education program (self-care)	12 sessions 5 min/session (telephone) 4 weeks Nurse	Adherence (compliance), IDWG, K, P, Alb, Hb, Hct, protein, cholesterol, transferrin	8	Yes	No
22	Kim et al. [38]	2014	Korea	1	N = 41 (E = 20, C = 21)	Age ≥ 19 years Mean age 58.9 (15.9) (E = 59.7 [16.4], C = 57.3 [14.9]) HD ≥ 1 month	Quasi-E	Education program (diet, video)	8 sessions 20~30 min/session (telephone) 8 weeks Nurse	Adherence (compliance), IDWG, K, P	7	Yes	No
23	Lee et al. [5]	2009	Korea	1	N = 41 (E = 22, C = 19)	Mean age (years) E = 58.6 (10.2), C = 56.5 (14.3) Mean HD duration E = 6.91 (4.62), C = 7.25 (4.18)	Quasi-E	Education &counseling program	18 sessions 20 min/session 6 weeks Nurse	Adherence (compliance), knowledge, IDWG, K, P	8	No	No
24	Seyyedrasooli et al. [39]	2013	Iran	1	N = 71 (E = 38, C = 33)	Age ≥ 18 years Mean age E = 47.5 (12.8), C = 48.1 (11.9) HD ≥ 6 months	RCT	Education program	6 sessions 45 min/session 6 weeks Nurse	Adherence (ESRD-AQ)	6	No	No
25	Hashemi et al. [40]	2018	Iran	1	N = 98 (E = 48, C = 50)	Mean age (years) E = 62.33 (14.22), C = 59.50 (16.14) Mean HD duration (months) E = 33.65 (33.13), C = 31.50 (30.22)	RCT	Education &counseling & training program	13~15 sessions 30~45 min/session 12 weeks Nurse	Adherence (ESRD-AQ)	9	No	No

Abbreviations: Alb, albumin; BDI-SF, the Beck Depression Inventory-Short Form; BP, blood pressure; BUN, blood urea nitrogen; C, control group; Ca, Calcium; CHeKS, Chronic Hemodialysis Knowledge Survey; Cr, Creatinine; DDFQ, The dialysis diet and fluid non-adherence questionnaire; DQOL, Diabetes Quality of Life; E, experimental group; E1, experimental group 1; E2, experimental group 2; ESRD, end-stage renal disease; ESRD-AQ, End-Stage Renal Disease Adherence Questionnaire; FCHPS, the Fluid Control in Hemodialysis Patients Scale; F/U, Follow up; GR-SMAQ-HD, Greek version of Simplified Medication Adherence Questionnaire for patients undergoing hemodialysis; Hb, hemoglobin; Hct, hematocrit; HD, hemodialysis; HRQoL, the health-related quality of life instrument; IDWG, Interdialytic Weight Gain; K, potassium; KDQ, Kidney Disease Questionnaire; Kt/V, a number used to quantify hemodialysis treatment adequacy; MMAS-4, The 4-item Morisky Medication Adherence Scale; MMAS-8, The 8-item Morisky Medication Adherence Scale; MOS, Medical Outcomes Study; MSPSS, Multidimensional Scale of Perceived Social Support; MVQoLI-15, the Missoula-VITAS Quality of Life Index; N, number of subjects; Na, sodium; P, phosphorus; PG-SGA, Patient-Generated Subjective Global Assessment; PSNCQQ, Patient Satisfaction with Nursing Care Quality Questionnaire; QoL, Quality of Life; Quasi-E, quasi-experimental studies; RCT, randomized controlled trials; SF-36, Short Form 36; SMS, short message service; UF, ultrafiltration; UK, United Kingdom; USA, United states of America; WHOQOL-SF: World Health Organization Quality of Life Assessment-Short Form Health Survey.

**Table 3 ijerph-19-11657-t003:** Quality assessment of the included studies.

**Joanna Briggs Institute Critical Appraisal Tool Checklist for Randomized Controlled Trials**														
**Study ID**	**Random Assignment**	**Allocation Concealment**	**Similarity between Groups**	**Blinding: Participants**	**Blinding: Delivering Treatment**	**Blinding: Outcome Assessors**	**Exposure to Similar Treatment**	**Follow-Up Completion**	**Intention-To-Treat Analysis**	**The Similarity in Measuring Outcomes**	**Reliability in Measuring Outcomes**	**Appropriate Statistical Analysis**	**Appropriate Trial Design**	**Total Score**
1	1	0	1	0	0	0	1	1	1	1	0	1	1	8
2	1	0	0	0	0	0	0	1	1	1	0	1	1	6
5	1	1	1	1	0	1	1	1	1	1	0	1	1	11
7	1	0	1	0	0	0	0	1	1	1	0	1	1	7
8	1	1	0	0	0	0	1	1	1	1	0	1	1	8
10	1	0	0	0	0	0	1	1	1	1	0	1	1	7
12	1	1	1	1	1	0	1	1	1	1	0	0	1	10
14	1	1	1	0	0	0	1	1	1	1	0	1	1	9
16	1	0	1	0	0	1	1	1	1	1	0	1	1	9
17	1	1	1	0	0	0	1	1	1	1	0	1	1	9
18	1	0	0	0	0	0	1	1	1	1	0	1	1	7
24	1	0	0	0	0	0	0	1	1	1	0	1	1	6
25	1	1	0	1	0	0	1	1	1	1	1	1	1	10
Subtotal	13	6	7	3	1	2	10	13	13	13	1	12	13	8.23
**Joanna Briggs Institute Critical Appraisal Tools Checklist for Quasi-Experimental Studies**														
**Study ID**	**Clarity of Cause and Outcome Effects**	**Similarity between Groups**	**Exposure to Similar Treatment**	**Comparison of the Treated Groups**	**Multiple Measurements**	**Follow-Up Completion**		**The Similarity in Measuring Outcomes**		**Reliability in Measuring Outcomes**		**Appropriate Statistical Analysis**		**Total Score**
3	1	1	1	1	1	1		1		0		1		8
4	1	0	0	1	1	1		1		0		1		6
6	1	1	0	1	1	1		1		1		1		8
9	1	1	0	1	1	1		1		1		1		8
11	1	1	0	1	1	1		1		1		1		8
13	1	1	1	1	1	1		1		0		1		8
15	1	1	0	1	1	1		1		0		1		7
19	1	1	0	1	1	1		1		1		1		8
20	1	1	0	1	1	0		1		1		1		7
21	1	1	0	1	1	1		1		1		1		8
22	1	0	0	1	1	1		1		1		1		7
23	1	1	0	1	1	1		1		1		1		8
Subtotal	12	10	2	12	12	11		12		8		12		7.58

**Table 4 ijerph-19-11657-t004:** Subgroup analysis regarding treatment adherence by study characteristics.

Characteristics	Subgroup	K	Study ID	N	Overall ES	95% CI		Z (*p*)	I^2^ (%)
						Lower Limit	Upper Limit		
Location (country of publication)	East Asia	12	3, 5, 6, 11, 13, 15, 16, 19, 20, 21, 22, 23	952	0.81	0.45	1.18	4.40 (<0.001)	86.1
	West Asia	8	1, 7, 8, 9, 10, 14, 24, 25	563	2.10	1.16	3.05	4.37 (<0.001)	95.5
	South Asia	2	2, 17	253	1.06	0.75	1.37	6.64 (<0.001)	22.3
	Others	3	4, 12, 18	257	0.26	–0.30	0.83	0.92 (0.360)	79.0
Study centers	1	19	1, 2, 3, 4, 6, 7, 8, 9, 10, 11, 14, 15, 16, 19, 20, 22, 23, 24, 25	1251	1.35	0.87	1.82	5.56 (<0.001)	92.9
	>1	6	5, 12, 13, 17, 18, 21	774	0.71	0.35	1.07	3.86 (<0.001)	82.9
Subjects	<70	11	1, 4, 6, 7, 8, 11, 14, 15, 19, 22, 23	592	1.12	0.69	1.56	5.10 (<0.001)	89.9
	≥70	14	2, 3, 5, 9, 10, 12, 13, 16, 17, 18, 20, 21, 24, 25	1433	1.10	0.67	1.53	5.02 (<0.001)	94.9
Study design	Quasi-E	12	3, 4, 6, 9, 11, 13, 15, 19, 20, 21, 22, 23	775	0.97	0.61	1.33	5.27 (<0.001)	83.4
	RCT	13	1, 2, 5, 7, 8, 10, 12, 14, 16, 17, 18, 24, 25	1250	1.27	0.75	1.80	4.74 (<0.001)	94.3
Interventions	Educational	20	1, 2, 3, 4, 7, 8, 9, 10, 11, 15, 16, 17, 18, 19, 20, 21, 22, 23, 24, 25	1496	1.23	0.83	1.63	5.98 (<0.001)	92.2
	Others	5	5, 6, 12, 13, 14	529	0.66	0.20	1.12	2.82 (0.005)	83.6
Sessions ^†^	≤12	19	1, 3, 4, 5, 6, 7, 9, 10b, 11, 13, 14, 15, 16, 18, 19, 20, 21, 22, 24	1460	0.82	0.53	1.1	1.60 (<0.001)	85.5
	>12	4	8, 17, 23, 25	358	2.94	1.36	4.51	3.65 (<0.001)	96.6
Follow-up	Yes	12	5, 7, 8, 11, 12, 13, 14, 16, 17, 19, 20, 23	1043	0.78	0.44	1.11	4.59 (<0.001)	84.1
	No	13	1, 2, 3, 4, 6, 9, 10, 15, 18, 21, 22, 24, 25	982	1.47	0.90	2.04	5.04 (<0.001)	94.0
Quality assessment score	Below the mean	11	1, 2, 4, 7, 8, 10, 15, 18, 20, 22, 24	796	0.79	0.40	1.18	3.97 (<0.001)	85.6
	Above the mean	14	3, 5, 6, 9, 11, 12, 13, 14, 16, 17, 19, 21, 23, 25	1229	1.34	0.85	1.82	5.40 (<0.001)	93.3
Funding	Yes	10	1, 5, 7, 10, 11, 13, 14, 18, 21, 22	863	0.68	0.37	0.98	4.28 (<0.001)	80.2
	No	15	2, 3, 4, 6, 8, 9, 12, 15, 16, 17, 19, 20, 23, 24, 25	1162	1.51	0.97	2.05	5.47 (<0.001)	93.8
Theory-based	Yes	6	3, 5, 6, 12, 13, 18	671	0.70	0.11	1.29	2.31 (0.021)	92.4
	No	19	1, 2, 4, 7, 8, 9, 10, 11, 14, 15, 16, 17, 19, 20, 21, 22, 23, 24, 25	1354	1.23	0.84	1.62	6.22 (<0.001)	90.7

^†^ Missing data: Below the mean, ≤8.2 (RCTs), ≤7.6 (quasi-experimental studies); Above the mean, ≥8.2 (RCTs), ≥7.6 (quasi-experimental studies). Abbreviations: CI, Confidence interval; ES, Effect size; K, Number of studies; Quasi-E, Quasi-experimental study; RCT, randomized controlled trials.

**Table 5 ijerph-19-11657-t005:** Meta-regression analysis evaluating treatment adherence.

Covariate (Ref.)	Estimate	SE	Z	*p*
Location (country of publication; Ref. = Others)	0.74	0.14	5.42	<0.001
Asia				
Study centers (Ref. > 1)	0.47	0.10	4.92	<0.001
1				
Participants (Ref. ≥ 70)	0.42	0.11	3.90	<0.001
<70				
Study design (Ref. = Quasi-E)	−0.21	0.10	−2.22	0.026
RCT				
Intervention (Ref. = Other)	0.37	0.10	3.52	<0.001
Educational				
Sessions (Ref. ≤ 12)	1.11	0.00	3.99	<0.001
>12				
Quality assessment score (Ref. = Below the mean)	0.23	0.10	2.43	0.015
Above the mean				
Funding (Ref. = No)	−0.48	0.10	−5.06	<0.001
Yes				
Theory-based (Ref. = No)	−0.48	0.10	−4.84	<0.001
Yes				

Quasi-E, Quasi-experimental study; RCT, randomized controlled trials; Ref. = reference; SE, standard error.

**Table 6 ijerph-19-11657-t006:** Effects of the evaluated interventions on secondary outcomes.

Variables	Number of Studies	N	Overall ES	95% CI		Z (*p*)	I^2^ (%)
				Lower Limit	Upper Limit		
IDWG	13	1186	–0.29	–0.52	–0.06	–2.48 (0.013)	73.4
P	10	800	−0.15	–0.37	0.07	–1.38 (0.170)	56
K	9	730	−0.24	−0.64	0.15	−1.20 (0.230)	85.2

Abbreviations: CI, Confidence interval; ES, Effect size; IDWG, Interdialytic weight gain; K, Potassium; P, Phosphorous.

**Table 7 ijerph-19-11657-t007:** Publication bias test evaluating intervention programs and treatment adherence.

Publication Bias Test		Coefficient	SE	95% CI		Z	*p*
				Lower Limit	Upper Limit		
Egger’s regression test	Intercept	7.46	1.69	4.15	10.76	4.42	<0.001
	Slope	−1.11	0.45	−1.98	−0.24	−2.49	0.013
		Tau-b		Ties		Z	*p*
Begg’s test	Standard	0.40		31		3.18	0.001
	Corrected	0.40		31		3.16	0.002
		Hg		95% CI			
				Lower limit		Upper limit	
Trim-and-fill	Original	1.10		0.77		1.43	
	Trim-and-fill	0.50		0.41		0.58	

CI, Confidence interval; Hg, Hedges’ g; SE, standard error.

## Data Availability

The secondary, previously published data used for this meta-analysis, though not available in a public repository, will be made available to other researchers upon reasonable request.

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
