# Peer review of "Effect of Treatment Adherence Improvement Program in Hemodialysis Patients: A Systematic Review and Meta-Analysis"

_ijerph, 2022, doi:10.3390/ijerph191811657_

Round 1

Reviewer 1 Report

This manuscript contains a thorough review and meta-analysis of hemodialysis patient treatment adherence improvement programs. Inadequate treatment adherence is a common problem among hemodialysis patients, causing acute and chronic problems as well as an increase in morbidity and mortality. A variety of research features were associated with larger synthesized effect sizes based on univariate meta-regression. When designing hemodialysis treatment adherence enhancement programs, this study provides evidence-based advice for improving program efficacy. Several points need to be revised in this section:

1.The first paragraph of INTRODUCTION should be summarized in two sentences to allow for the actual problem description.

2. The PRISMA Flow Diagram in Figure 1 is unclear; another image should be used instead.

3.In Figure 2, the scale of standardised mean difference suggests that adjustments be made.

4.The rationale for selecting primary and secondary outcomes

5.Is there anything the authors can do to improve the publication bias?

Reviewer 2 Report

The systematic review and meta-analysis by Kim et al. evaluates the effects of treatment adherence enhancement programs on treatment adherence and secondary outcomes for hemodialysis patients. The review could be of interest to the readers of IJERPH, and the quality of the data give interesting results and make it a practical tool for the physician. Moreover, the topic is not a new one but it is still a matter of the current debate.

 Some points need to be addressed:

1.       The English language has to be largely revisited

2.       I cannot really comment on the strategy used to identify the studies but per the Authors explanation it seems to be really good. The Authors included the randomized controlled trials (RCTs) and quasi-experimental studies. It is pit that some studies were eliminated due to the observational nature of the studies, which would have conferred a view of real life.

3.       As for the analysis:

- I agree with the approach for the RCTs. It is fine for the RCT studies but must be said explicitly in the methods as why the synthesis was not conducted. It is not that obvious for all readers of IJERPH.
- The meta-analysis how was performed? This important aspect has to be further deeped

- Figure 2. The forrest plot is very unclear. The small-squares are hard to understand.

- Figure 3. The caption to Figure is not complete, please clarify

4.       Results:
- No comments in general
- There are some typos

- Please upload figures with better resolution. It is quite hard for my eyes to read this.

Discussion
- The message is clearly driven to the reader

Author Response

September 11, 2022

Prof. Dr. Paul B. Tchounwou

Editor-in-Chief

Dear Editor:

We wish to re-submit the manuscript titled “Effect of Treatment Adherence Improvement Program in Hemodialysis Patients: A Systematic Review and Meta-analysis”. The manuscript ID is ijerph-1886886.

We thank you and the reviewers for your constructive comments on our manuscript. It is with great pleasure that we resubmit our article for further consideration. We have incorporated changes that reflect the detailed suggestions and comments you and the reviewers have graciously provided. We hope that our edits and the responses we have provided satisfactorily address all the issues and concerns you and the reviewers have noted. The changes in the revised manuscript are in red text for ease of identification, and the responses to all comments have been attached.

Thank you for your consideration. We look forward to hearing from you and would be happy to make further changes, if required.

Sincerely,

Mi-Kyoung Cho

Department of Nursing Science, School of Medicine

Chungbuk National University, Cheongju 28644, Korea

 Email: ciamkcho@gmail.com

Tel.: +82-43-249-1797

Round 2

Reviewer 2 Report

I have no more comments to the Authors. The paper can be accepted in the present form